# *Sphingomonas sediminicola* Dae20 Is a Highly Promising Beneficial Bacteria for Crop Biostimulation Due to Its Positive Effects on Plant Growth and Development

**DOI:** 10.3390/microorganisms11082061

**Published:** 2023-08-11

**Authors:** Candice Mazoyon, Manuella Catterou, Abdelrahman Alahmad, Gaëlle Mongelard, Stéphanie Guénin, Vivien Sarazin, Fréderic Dubois, Jérôme Duclercq

**Affiliations:** 1Ecologie et Dynamique des Systèmes Anthropisés (EDYSAN, UMR7058 CNRS), Université de Picardie Jules Verne (UPJV), 80000 Amiens, France; candice.mazoyon@outlook.fr (C.M.); manuella.catterou@u-picardie.fr (M.C.); abdel-rahman.alahmad@unilasalle.fr (A.A.); frederic.dubois@u-picardie.fr (F.D.); 2Agroécologie, Hydrogéochimie, Milieux et Ressources (AGHYLE, UP2018.C101) UniLaSalle Rouen, SFR NORVEGE FED 4277, 76130 Mont-Saint Aignan, France; 3Centre de Ressources Régionales en Biologie Moléculaire (CRRBM), Université de Picardie Jules Verne (UPJV), 80000 Amiens, France; gaelle.mongelard@u-picardie.fr (G.M.); stephanie.vandecasteele@u-picardie.fr (S.G.); 4AgroStation, 68700 Aspach-le-Bas, France; sarazinv@agrostation.fr

**Keywords:** plant growth-promoting bacteria (PGPR), *Sphingomonas sediminicola* Dae20, root stimulation, growth promotion, *Brassicaceae* plants, sustainable agriculture

## Abstract

Current agricultural practices rely heavily on synthetic fertilizers, which not only consume a lot of energy but also disrupt the ecological balance. The overuse of synthetic fertilizers has led to soil degradation. In a more sustainable approach, alternative methods based on biological interactions, such as plant growth-promoting bacteria (PGPRs), are being explored. PGPRs, which include both symbiotic and free-living bacteria, form mutualistic relationships with plants by enhancing nutrient availability, producing growth regulators, and regulating stress responses. This study investigated the potential of *Sphingomonas sediminicola* Dae20, an α-Proteobacteria species commonly found in the rhizosphere, as a beneficial PGPR. We observed that *S. sediminicola* Dae20 stimulated the root system and growth of three different plant species in the *Brassicaceae* family, including *Arabidopsis thaliana*, mustard, and rapeseed. The bacterium produced auxin, nitric oxide, siderophores and showed ACC deaminase activity. In addition to activating an auxin response in the plant, *S. sediminicola* Dae20 exhibited the ability to modulate other plant hormones, such as abscisic acid, jasmonic acid and salicylic acid, which are critical for plant development and defense responses. This study highlights the multifunctional properties of *S. sediminicola* Dae20 as a promising PGPR and underscores the importance of identifying effective and versatile beneficial bacteria to improve plant nutrition and promote sustainable agricultural practices.

## 1. Introduction

Current agricultural practices use large amounts of synthetic fertilizers to improve plant nutrition. However, synthetic fertilization is energy intensive and disrupts the balance of the ecosystem. Excessive use of synthetic fertilizers has severely degraded soil quality both chemically and biologically [1]. Alternative approaches are being envisaged to provide nutrients to plants and maintain ecosystem stability. These include the development of agricultural practices based on biological interactions, particularly those involving rhizospheric soil bacteria [2,3,4]. These microorganisms, referred to as plant growth-promoting bacteria (PGPR), are attracted to plant roots that release organic nutrients under optimal or stressful conditions [2,5]. Bacterial populations are able to metabolize exudates and proliferate in the rhizosphere, providing nutrients important for plant growth and yield [6,7]. This nutrient exchange between plant and bacteria, also known as a mutualistic relationship, occurs in symbiosis with the host plant or as free bacteria in the soil. Symbiotic PGPR, called rhizobia, form a symbiotic relationship with their legumes by forming nodules on plant roots in which the plant receives nitrogen from the symbiont in exchange for carbon [8]. Free-living PGPRs, such as *Azospirillum*, *Bacillus*, *Enterobacter*, *Klebsiella*, and *Pseudomonas*, also affect nitrogen fixation, phosphate solubilization, iron availability, and can also produce various growth regulators and siderophores, or modulate stress hormone levels in plants [9]. Depending on the plant species, PGPRs may employ one or more of these mechanisms and are of particular interest when they combine multiple beneficial mechanisms for different plant species [10]. *Rhizobium leguminosarum*, for example, is a symbiotic PGPR than can fix atmospheric nitrogen in pea nodules, and a free-living PGPR in the soil that can stimulate rice growth by producing auxin and lowering plant ethylene levels, which is known to inhibit plant growth through its aminocyclopropane carboxylate (ACC) deaminase activity [11]. It is known that the production of other bacterial substances and enzymatic activities promote plant growth. Nitric oxide production by PGPRs, such as *Azospirillum brasilense*, improves plant growth [10]. Similarly, some PGPRs, such as *Pseudomonas fluorescens*, *Rhizobium leguminosarum*, and *Bacillus simplex*, are able to enhance the uptake of iron, an essential element for plant development through the production of iron-chelating siderophores [10]. PGPRs not only promote plant growth but could also trigger a mechanism called induced systemic resistance (ISR) that enhances plant protection against various diseases. The ISR mechanism is supported by plant growth regulators such as salicylic acid (SA), jasmonic acid (JA) and ethylene (ET) as they play a crucial role in this process [12,13].

Several PGPR species are now used in bio-formulations to stimulate the growth of various plants. Most bacterial bio-stimulants consist of a limited number of recurrent PGPRs, such as *Azospirillum*, *Azotobacter*, *Bacillus*, *Pseudomonas* and *Rhizobium*, with each combination being specific to the plant species [14]. It is important to identify more effective and multifunctional beneficial PGPRs to maximize the number of PGPRs that can fertilize the rhizosphere of plants. To determine whether a bacterial species is a PGPR, various plant traits such as plant height, root length, lateral root density, root hair length and density, and plant biomass are usually used as phenotypic markers [15].

*Sphingomonas* are α-Proteobacteria found in the rhizosphere and are generally described to promote plant growth [16,17], with some of them even producing plant growth regulators such as auxin and gibberellin [18,19]. In a recent study, we showed that the species *Sphingomonas sediminicola* Dae20 is able to improve nitrogen supply to peas by developing functional nodules when inoculated with plants [20]. Building on this premise, our study seeks to explore the broader applicability of *S. sediminicola* Dae20 as a PGPR. Our primary objective is to evaluate whether *S. sediminicola* Dae20 exhibits the characteristic features of effective PGPRs in a broader range of plant species, particularly in the *Brassicaceae* family. We investigate the prospect that *S. sediminicola* Dae20 effectively stimulates root system development and overall growth of several *Brassicaceae* species. Our investigation includes a comprehensive analysis of several PGPR properties, including auxin production, nitric oxide emission, siderophore synthesis, and ACC deaminase activity. In addition, we investigate the transcriptional changes induced by *S. sediminicola* Dae20 inoculation, focusing on genes related to hormonal signaling pathways. In a broader context, our study aims to disclose the potential of *S. sediminicola* Dae20 as a robust and multifunctional PGPR and to contribute to the available tools for sustainable agricultural practices. By examining the interplay of *S. sediminicola* Dae20 with various *Brassicaceae* species, we aim to demonstrate the broader implications of PGPR-based strategies to promote resilient and productive agricultural systems.

## 2. Materials and Methods

### 2.1. Bacterial Cultures

The bacterial strain used in this study was *Sphingomonas sediminicola* Dae20 [20], provided by the Leibniz Institute DSMZ (DSM-18106). *S. sediminicola* Dae20 was grown in Reasoner’s 2A (R2A) medium (0.05% (*w*/*v*) yeast extract, 0.05% (*w*/*v*) peptone, 0.05% (*w*/*v*) casamino acids, 0.05% (*w*/*v*) glucose, 0.05% (*w*/*v*) starch, 0.03% (*w*/*v*) sodium pyruvate, 0.03% (*w*/*v*) K_2_HPO_4_, 0.005% (*w*/*v*) MgSO_4_, pH 7) for 72 h at 30 °C with 150 rpm constant shaking (SI600, STUART, Staffordshire, UK). Bacterial concentration expressed in number of colony-forming units (CFU) by mL was determined on R2A 1.5% (*w*/*v*) agar plates (VWR, Fontenay-sous-Bois, France) by the spiral method using an EasySpiral^®^ automatic plater (Intersciences, Mourjou, France) and counted using a colony counter Scan^®^500 (Interscience).

### 2.2. In Vitro Arabidopsis Assay

*Arabidopsis thaliana* ecotype Columbia (Col-0) was obtained from the Nottingham Arabidopsis Stock Centre (NASC). The following GUS reporter lines were also used in this study: *DR5::GUS* and *ARR5::GUS* [21]. *Arabidopsis* seeds were surface-sterilized with ethanol and cold-stratified for 48 h before being sown in square Petri dishes (VWR, Fontenay-sous-Bois, France) containing *Arabidopsis* medium (AM, 0.23% (*w*/*v*) Murashige and Skoog salts, 0.05% (*w*/*v*) myo-inositol, 2% (*w*/*v*) sucrose, 0.8% (*w*/*v*) agar (Duchefa, Haarlem, The Netherlands), pH 5.8, stored at 4 °C for 2 days, and grown on vertically oriented plates in growth chambers under a 16 h light/8 h dark photoperiod at 21 °C.

For the phenotypic analysis, surface-sterilized and cold-stratified Col-0 seeds were sown in AM plates equally spaced and 2 cm from the top of the plate. In vitro inoculation of *A. thaliana* plants with *S. sediminicola* Dae20 was performed by replacing a strip of AM medium (6 cm^3^ = 12 × 1 × 0.5, L × l × H) 3 cm from the bottom of the dish with a similar strip of R2A medium containing 0 (non-inoculated condition), 5 × 10^4^, 5 × 10^8^ or 2 × 10^9^ CFU mL^−1^. Ten days after germination, some of the plates were used for the phenotypic study, while the others were used for a transcriptomic study.

For the phenotypic analysis, the plates were photographed with a Canon EF 100 mm camera (Canon). Root hairs (RHs) were recorded 1 cm from the root tip and imaged using a Zeiss SteREO Discovery V20 microscope (Carl-Zeiss, Goettingen, Germany). Then, the plant material was cleared [22] by incubating it at 65 °C for 10 min in a solution of 4% HCl and 20% methanol, followed by a 10 min incubation in 7% NaOH/60% ethanol at room temperature. Then, the seedlings were rehydrated by incubating them successively in 60, 40, 20, and 10% ethanol for 15 min, followed by a 30 min incubation in a solution of 25% glycerol and 5% ethanol. Finally, the material was placed in 50% glycerol. The lengths of the main root and of RHs were measured using ImageJ (v1.53v, http://rsb.info.nih.gov/, accessed on 1 December 2022). Lateral roots (LRs) were counted using a differential interference contrast (DIC) microscope (Nikon Eclipse E800, Nikon, Tokyo, Japan), and the total number of LRs per cm of root length was evaluated. Dry shoot and root biomass were also measured.

To determine whether the effect of *S. sediminicola* Dae20 was caused solely by its volatile organic compounds (VOCs), surface-sterilized and cold-stratified Col-0 seeds were sown in AM plates using the same procedure as above. In vitro inoculation of these plants with *S. sediminicola* Dae20 was performed by replacing a strip of AM medium with a similar strip of R2A medium containing 0 (non-inoculated condition) or 2 × 10^9^ CFU mL^−1^. An additional strip, 1 cm wide, was removed above the R2A medium strip and left free to prevent diffusion of soluble compounds. Ten days after germination, the plates were photographed. After clearing, seedlings were mounted in 50% glycerol and LRs counted, and the total number of LRs per cm of root length was determined. Fresh shoot and root biomass were also measured.

Surface-sterilized and cold-stratified *DR5::GUS* and *ARR5::GUS* transgenic lines seeds were sown in AM plates following the same procedure as above. In vitro inoculation of these plants with *S. sediminicola* Dae20 was performed by replacing a strip of AM medium with a similar strip of R2A medium containing 0 (non-inoculated condition) or 2 × 10^9^ CFU mL^−1^. Ten days after germination, the seedlings were incubated in the dark at 37 °C in a reaction buffer containing 100 mM sodium phosphate buffer (pH 7) with 0.01% (*v*/*v*) Triton X-100, 5 mM ferroferricyanide buffer and 1 mg.mL^−1^ X-Gluc (5-bromo-4-chloro-3-indolyl-beta-D-glucuronic acid, cyclohexylammonium salt) as a chromogenic substrate. At least 20 seedlings per condition were analyzed and experiments were repeated three times independently. After clearing, the GUS signal was observed using the DIC microscope.

For the transcriptomic analysis, total RNAs of whole plants were extracted with the RNeasy Plant Mini Kit (Qiagen, Hilden, Germany). A DNase treatment with the RNAse-free DNAse kit (Qiagen, Hilden, Germany) was performed for 15 min at 25 °C. The cDNA was prepared from 500 ng RNA with the iScript cDNA synthesis kit (Biorad, Hercules, CA, USA) according to the manufacturer’s instructions, and it was analyzed on a LightCycler 480 (Roche Diagnostics, Penzberg, Germany) with the SYBR Green I Master kit (Roche Diagnostics, Penzberg, Germany). Genes involved in auxin, abscisic acid, jasmonic acid and salicylic acid pathways were amplified by PCR in 384-well optical reaction plates heated for 10 min at 94 °C to activate the hot-start Taq DNA polymerase, followed by 30 cycles of denaturation for 60 s at 95 °C and annealing/extension for 60 s at 55 °C. Targets were quantified with specific primer pairs designed with QuantPrime [23] (Appendix A). Expression levels were normalized with an actin reference gene, *ACT2* (Appendix A). Relative gene expression was estimated by the 2^-ΔΔ^Ct method transformed into gene expression factor difference [24].

### 2.3. Plant Growth Promoting Traits of S. sediminicola Dae20

To demonstrate the PGPR properties of *S. sediminicola* Dae20, we used bacterial cultures with a CFU density of 2 × 10^8^ CFU.

Indole acetic acid (IAA) production was determined from 1 mL cell-free supernatant of *S. sediminicola* Dae20 mixed with 100 μL orthophosphoric acid and 2 mL Salkowski reagent (50 mL, 35% perchloric acid, 1 mL 0.5 M FeCl_3_ solution). IAA production was measured spectrophotometrically at 530 nm and estimated against a standard curve of IAA (10–100 μg mL^−1^) [25].

Nitric oxide production was determined according to the colorimetric Griess assay [26]. One milliliter of cell-free supernatant of *S. sediminicola* Dae20 or a standard solution of sodium nitrite (0–200 μM) was mixed with an equal volume of Griess reagent (0.2% (*w*/*v*) naphthyl ethylene diamine and 2% (*w*/*v*) sulfanilamide in 5% (*v*/*v*) phosphoric acid). Nitric oxide reacts with Griess reagent to form a stable product that can be detected by its absorption at 540 nm.

Siderophores were quantified according to the chrome azurol S (CAS) liquid assay [27]. Briefly, 0.5 mL supernatant of *S. sediminicola* Dae20 culture was mixed with 0.5 mL CAS reagent. After 20 min, siderophore production was measured by absorbance at 630 nm. The percentage of siderophore unit (psu) was calculated as follows: [(Ar − As)/Ar] × 100 = % of siderophore units [28]. Here, Ar represents the reference absorbance (CAS solution and uninoculated broth) and As represents the absorbance of the sample (CAS solution and supernatant of the sample) [28].

The ACC deaminase activity of *S. sediminicola* Dae20 was measured according to Penrose and Glick [29]. Bacteria were seeded on DF medium [30] supplemented with 3 mM ACC or 0.2% (*w*/*v*) ammonium sulfate as the sole nitrogen source [31].

A phosphate solubilization test was performed using Pikovskaya’s agar medium [32].

### 2.4. Brassicaceae Assays

White mustard (*Sinapis alba*) cv. Ascot (Jardiland, Paris, France) and rapeseed (*Brassica napus* L.) cv. Ambassador (Limagrain Europe) seeds were surface-sterilized with a 3.5% (*v*/*v*) bleach solution and cold-stratified for 48 h. Five sterile and stratified seeds of mustard or rapeseed were sown in AM plates equally spaced at 2 cm from the top of the plate. In vitro inoculation with *S. sediminicola* Dae20 was performed by replacing a strip of AM medium 3 cm from the bottom of the dish by a similar strip of R2A medium containing 0 or 2 × 10^9^ CFU mL^−1^. Seven days after germination, RHs were observed and recorded at 1 cm from the root tip (arrows) using a Zeiss SteREO Discovery V20 microscope (Carl-Zeiss, Goettingen, Germany), and the length of the RHs were measured using ImageJ. At least 20 seedlings per condition were analyzed and experiments were repeated three times independently.

Surface-sterilized rapeseed and mustard, cold-stratified for 48 h, were also sown in the dark on a 1% (*w*/*v*) agar medium at 21 °C. Five days after germination, etiolated seedlings were transferred to a Quickpot QP 6 T/20 system (Puteaux, Limas, Villefranche-sur-Saône, France) containing 500 g sterile potting soil (Horticole Fal, NPK = 4-4-4; Puteaux). Plants were grown with a 16 h light/8 h dark photoperiod at 21 °C with a light intensity of 400 µmol m^−2^ s^−1^ and inoculated with 1 mL of *S. sediminicola* Dae20 growth culture containing 2 × 10^9^ CFU. For the non-inoculated condition, 1 mL of R2A medium was added to the seedlings. Plants were grown 30 days after inoculation and irrigated every 3 to 4 days with sterile distilled water.

Each plant was photographed, and plant height and root length were determined using ImageJ. Dry biomass of the above- and below-ground parts was measured after fresh parts were dried in an oven at 60 °C until a stable weight was reached. A relative measurement of chlorophyll content in the cotyledons as well as in the penultimate leaf was performed using a SPAD chlorophyll meter (SPAD-502Plus, Konica Minolta Optics, Osaka, Japan). SPAD values range from 0 to 100, with higher values indicating higher chlorophyll content in the leaves. The experiment was repeated at least three times independently, each performed with 10 to 20 plants by modality.

### 2.5. Statistical Analysis

Data were compared between treatments using a non-parametric Kruskal–Wallis one-way analysis of variance, followed by an ANOVA *post hoc* test if significant. All *p* values were adjusted using the Benjamini–Hochberg FDR procedure [33].

## 3. Results

### 3.1. S. sediminicola Dae20 Stimulated the Root System of A. thaliana

After 10 days of co-cultivation with *S. sediminicola* Dae20, we observed various changes in both roots and shoots during the development of *A. thaliana* (Figure 1). The presence of *S. sediminicola* Dae20 resulted in increased density of lateral roots (LRs) (Figure 1a) without altering the length of the primary root (Figure 1b). The number of LRs improved with increasing concentration of bacteria (Figure 1c). Both RH density and RH length were also increased in the presence of *S. sediminicola* Dae20 (Figure 1d,e). Altogether, this resulted in an increase in root (Figure 1f) and shoot biomass production compared to the non-inoculated plants (Figure 1g). For all measured parameters, inoculation with 2 × 10^9^ CFU mL^−1^ *S. sediminicola* Dae20 resulted in the highest values and is used below as the inoculation concentration of the bacteria.

As many bacteria such as *S. sediminicola* Dae20 can produce both soluble [34] and volatile organic compounds (VOCs) [35], we wanted to know the contribution of VOCs to the plant phenotypic changes induced by *S. sediminicola* Dae20. In the presence of only these VOCs, there was no detectable change in primary root growth or LR formation (Figure 2a,b) compared to non-inoculated plants. In addition, an increase in the fresh weight of both shoot and root components was observed when VOCs were applied (Figure 2c,d).

### 3.2. S. sediminicola Dae20 Has Several PGPR Characteristics

In the supernatant of the free-living *S. sediminicola* Dae20 culture, IAA was detected at a concentration of 1.25 ± 0.04 µg per mg dry weight (DW) of bacterial cells (Table 1).

This level was consistent with the increase and decrease in the GUS signal in the roots of 10-day-old seedlings *DR5::GUS* and A*RR5::GUS* seedlings, respectively (Figure 3a,b). In addition, we observed the development of *S. sediminicola* Dae20 on medium containing ACC as the only nitrogen source, indicating that the bacteria have ACC deaminase activity (Figure 3c). We also detected nitric oxide and siderophores in this supernatant (Table 1). In contrast, no development was observed on the Pikovskaya medium, indicating that *S. sediminicola* Dae20 did not have phosphate-solubilizing activity.

Moreover, we detected an increase in the transcripts of several genes involved the biosynthesis of auxin (*GH3.5*, *ARF7* and *ARF19*), abscisic acid (*ABI1*), jasmonic acid (*JAZ1*) and salicylic acid (*MYC2*, *PR1* and *EDS1*) in plants inoculated with *S. sediminicola* Dae20 (Figure 4).

### 3.3. S. sediminicola Dae20 Enhances the Growth and Root System of Brassicaceae Plants

As in *Arabidopsis*, the development of the root system of mustard or rapeseed seedlings was also influenced by the presence of *S. sediminicola* Dae20. Indeed, the production of soluble and volatile organic compounds by the bacteria during the 7-day growth period stimulated the presence and elongation of root hairs in rapeseed and mustard (Figure 5a–c).

Inoculation of mustard plants with *S. sediminicola* Dae20 resulted in taller plants (Figure 6a) with a longer root system (Figure 6b) and greater biomass compared to non-inoculated plants (Figure 6c). The SPAD values of cotyledons and penultimate leaves of inoculated mustard were also higher than those of non-inoculated plants (Figure 6d). Inoculation of *S. sediminicola* Dae20 on rapeseed plants also resulted in taller plants with longer root systems (Figure 6e,f). However, the shoot biomass of these plants was lower than that of the non-inoculated plants (Figure 6g). In addition, no differences were observed in root biomass or SPAD values (Figure 6h).

## 4. Discussion

Identifying new bacterial strains may be a critical step in the transition process towards a more sustainable agriculture by reducing the use of chemical inputs. Beneficial bacteria can help reduce reliance on chemical fertilizers, pesticides, and herbicides [36,37]. They can provide plants with the nutrients and protection they need for healthy growth and resistance to disease and environmental stress. In addition, the use of beneficial bacterial strains can improve soil quality and biodiversity, which promotes a more holistic approach to agricultural land management [38]. Although known strains such as *Rhizobium* sp., *Pseudomonas* sp., or *Bacillus* sp. can be effective [9,10], new strains should be sought, as certain strains may be more suitable for certain crops or conditions. Exploration of bacterial diversity can lead to the discovery of unique strains with special capabilities that improve crop productivity and promote sustainable agriculture. Furthermore, revealing new bacterial strains can broaden the range of options available for crop management and enhance the ability to adapt to climate and environmental changes.

Within this bacterial diversity, there are many candidates that can positively influence plant development. This is the case with *Sphingomonas* sp., which are usually identified in taxonomic inventory approaches using metabarcoding with abundance that can reach 40% of the total bacterial abundance [39,40,41,42]. As previously reported, *S. sediminicola* was found to be highly abundant in agricultural soils after pea culture [41]. This occurrence was found to be non-random, as the pea plant adjusts the composition of its root exudates to specifically attract this bacterium [34]. Once recruited, this bacterium promotes pea growth, induces the formation of root nodules, and fixes atmospheric nitrogen [20]. Here, we showed that *S. sediminicola* Dae20 can also promote root growth in three *Brassicaceae* species, *Arabidopsis*, mustard, and rapeseed. Other *Sphingomonas* strains, such as *S. Cra 20*, *S*. *SaMR12*, and *S*. *azotifigens*, have also been reported to promote the growth of *Brassicaceae* plants [40,43,44,45]. The effect of *S. sediminicola* Dae20 on the root system, characterized by a strong increase in branching and root hairs, is closely related to its ability to influence the hormonal balance of the plant, either through the direct production of auxin in the form of IAA, through the modulation of ethylene levels, or through the production of nitric oxide. Indeed, these molecules are encoded by genes present in the genome of the bacteria [20] and they are known to influence these processes in plants [46,47,48]. Several species of *Sphingomonas* have been described with these abilities, such as *S. LK11*, *S. paucimobilis*, *S. pokkalii* [18,49,50,51,52,53,54]. Stimulation of the root system by *S. sediminicola* Dae20 allows the plant to develop a more extensive root exchange zone, enabling it to take up nutrients and water more efficiently and to interact with other microorganisms to a greater extent. This enhanced root system could have significant effects on the nutrient balance of the plant, potentially leading to increased nutrient uptake and utilization. Therefore, it would be interesting to evaluate the nutrient balance of these plants to determine the extent to which *S. sediminicola* Dae20 and other *Sphingomonas* species can improve plant growth and development. Moreover, extending this assessment to grain yield and composition could provide a more holistic perspective, especially for crops of this type. By examining not only vegetative traits, but also reproductive outcomes and grain nutritional quality, we can decipher the full impact of *S. sediminicola* Dae20 on the entire life cycle of these crops.

Several *Sphingomonas* species have been described in the literature as being capable of producing siderophores, including *S. LK11*, *S. paucimobilis*, and *S. pokkalii* [18,51,54]. Our research has also shown that *S. sediminicola* Dae20 is able to produce siderophores and it is consistent with the genes involved in iron absorption, storage and transport trapping found in the genome of *S. sediminicola* Dae20 [20]. By producing siderophores, *S. sediminicola* Dae20 may potentially enhance the ability of plants to acquire iron, which could lead to improved growth and development. Therefore, it would be interesting to evaluate the efficacy of this bacterium in the context of plants developing under iron deficiency. Further research is needed to investigate the role of *S. sediminicola* Dae20 and other siderophore-producing *Sphingomonas* species in enhancing plant growth under iron deficiency, and to evaluate the potential for these bacteria to be used as biofertilizers or bioinoculants for crop production.

*S. sediminicola* Dae20 not only has beneficial effects on plant growth but can also stimulate genes involved in plant hormone synthesis and signaling pathways, such as salicylic acid, jasmonic acid, and ethylene. These signaling pathways are associated with induced systemic resistance (ISR), a mechanism by which plants activate their defense responses against pathogenic microorganisms [55]. *Sphingomonas* bacteria not only promote plant growth, but also increase plant resistance to pathogens. Different species of *Sphingomonas* have been shown to improve plant response to a variety of pathogens [56], including fungi and bacteria. For example, *S. sp. 27* has been found to increase tolerance to Fusarium wilt, a serious fungal disease that affects many crops [57]. Similarly, *S. trueperi* has been shown to help rice respond better to *Achlya klebsiana* and *Pythium spinosum*, two pathogens that cause severe root rot in rice plants [58]. Therefore, it would be interesting to investigate whether plants that benefit from the positive effects of *S. sediminicola* Dae20 also respond better to pathogens.

In addition, *S. sediminicola* Dae20, like many PGPRs, promotes root growth and improves nutrient uptake by plants. As a result, plants inoculated with these bacteria also have higher chlorophyll content, leading to improved photosynthesis, stronger growth, and increased resistance to stressful environments [59,60,61]. Some *Sphingomonas* species, such as *S. Cra20*, *S. LK11*, and *S. wittichii* RW1 [44,53,62], have been reported to enhance plant response to drought. Moreover, *S. paucimobilis* ZJSH1 and *S. nostoxanthinifaciens* have shown their potential in saline environments [63,64]. Therefore, it would be of interest to evaluate the effects of *S. sediminicola* Dae20 under these stress conditions.

The influence of *Sphingomonas sediminicola* Dae20 on root architecture observed in in vitro culture was further confirmed under controlled conditions with potting soil, with the exception of rapeseed. The differential response in *Brassicaceae* species, such as rapeseed, highlights the complex nature of plant–microbe interactions and the specificity of these relationships. It is plausible that rapeseed has specific physiological or genetic characteristics that influence its interaction with *S. sediminicola* Dae20 and result in a different response than other plant species studied. Soil composition, in combination with plant characteristics, may play a role in mediating these effects. In addition, bacteria recruitment by root exudates could contribute to the species-specific responses observed [65]. Investigating the detailed composition of root exudates could provide valuable insights into the compatibility between *S. sediminicola* Dae20 and different plant species. Understanding the specific factors contributing to the interaction between *S. sediminicola* Dae20 and plant roots will allow us to optimize the use of this bacterium in agriculture and promote sustainable plant–microbe associations.

## 5. Conclusions

Considering our previous results, *S. sediminicola* Dae20 emerges as a novel tool for improving agricultural practices. With a demonstrated ability to increase legume productivity regardless of soil nitrogen content, this bacterium offers a promising alternative to conventional rhizobial inoculants, especially in soils that are not nitrogen poor [20]. The significant abundance of *S. sediminicola* Dae20 in agricultural soils following pea cultivation is not a coincidence, but rather a strategic preference of the crop [34,41]. This strategic choice highlights the potential role of *S. sediminicola* Dae20 in shaping soil microbial communities, thus influencing nutrient cycling and availability.

Importantly, our studies reveal a broader spectrum of the influence of *S. sediminicola* Dae20. Its beneficial effects are not limited to legumes or its symbiotic nitrogen-fixing abilities [20]. In fact, this bacterium shows its potential to enhance major traits in several plant species, including the *Brassicaceae* family, as demonstrated by this study.

As we enter an era of sustainable agriculture, the role of *S. sediminicola* Dae20 as a versatile ally becomes clear. It demonstrates the power of the microbial biointrant in optimizing crop production while respecting ecological subtleties. As we deepen our knowledge of its interactions with various plants and decipher the mechanisms behind its beneficial effects, we will be able to harness its potential for a more resilient and productive agricultural future.

In summary, our study makes an important contribution to the complex puzzle of sustainable agriculture by positioning *S. sediminicola* Dae20 as an effective means to increase plant growth and agricultural productivity beyond traditional limits. As the challenges of feeding a growing global population continue to increase, the potential of such innovative solutions becomes increasingly important.

## Figures and Tables

**Figure 1 microorganisms-11-02061-f001:**
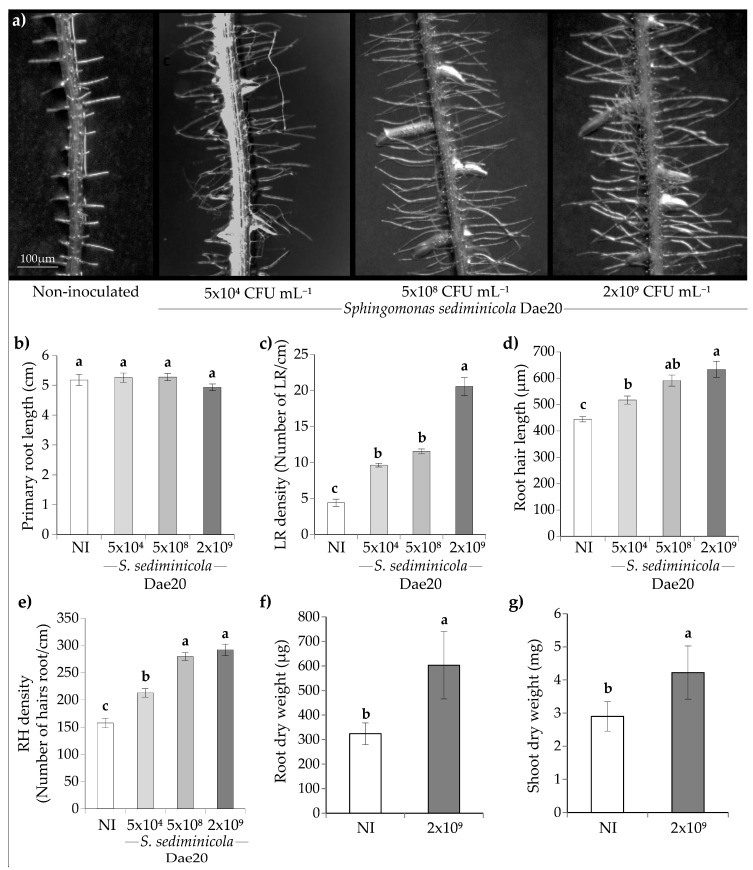
*Arabidopsis thaliana* root phenotype (**a**), primary root length (**b**), lateral root (LR) density (**c**), root hair (RH) length (**d**), RH density (**e**), shoot dry weight (**f**) and root dry weight (**g**) of 10-day-old *Arabidopsis thaliana* not inoculated (NI) or inoculated with different bacterial concentrations expressed as the number of colony-forming units (CFU) of *S. sediminicola* Dae20. Error bars represent SD. Letters represent a significant difference among modalities.

**Figure 2 microorganisms-11-02061-f002:**
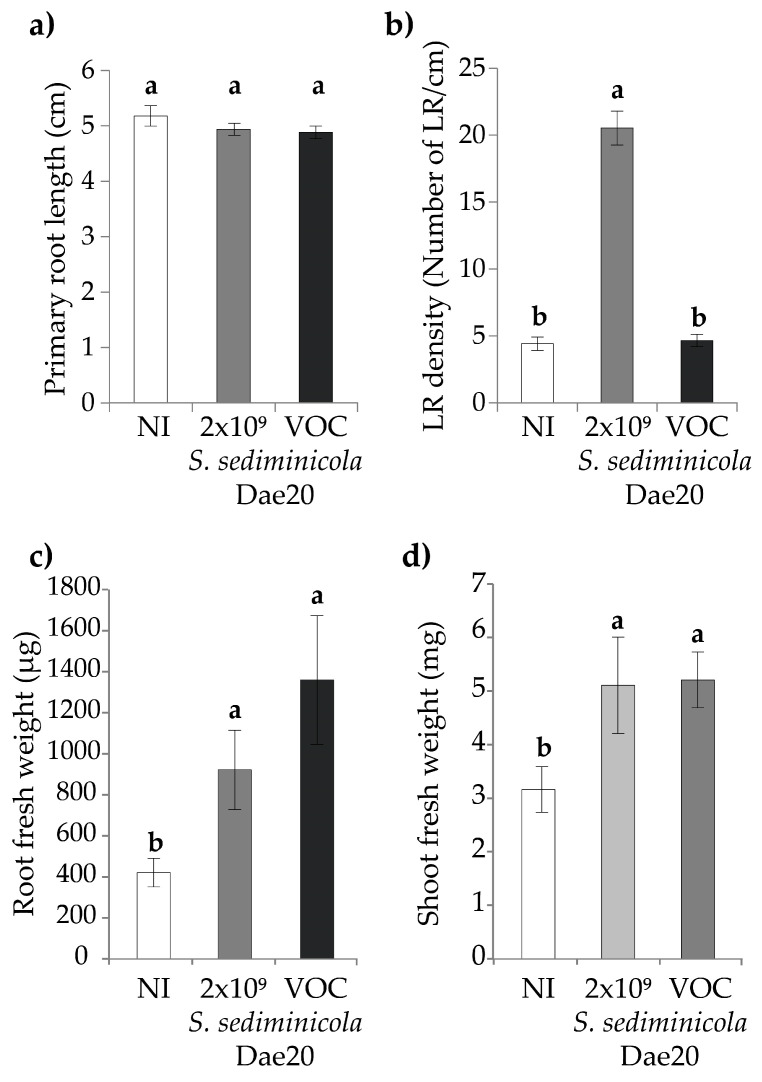
(**a**) Primary root, (**b**) lateral root (LR) density, (**c**) shoot fresh weight and (**d**) root fresh weight in 10-day-old *Arabidopsis thaliana* not inoculated (NI), inoculated with *S. sediminicola* Dae20 or grown in the presence of volatile organic compounds (VOC) released by the bacterial strain. Error bars represent SD. Letters represent a significant difference among modalities.

**Figure 3 microorganisms-11-02061-f003:**
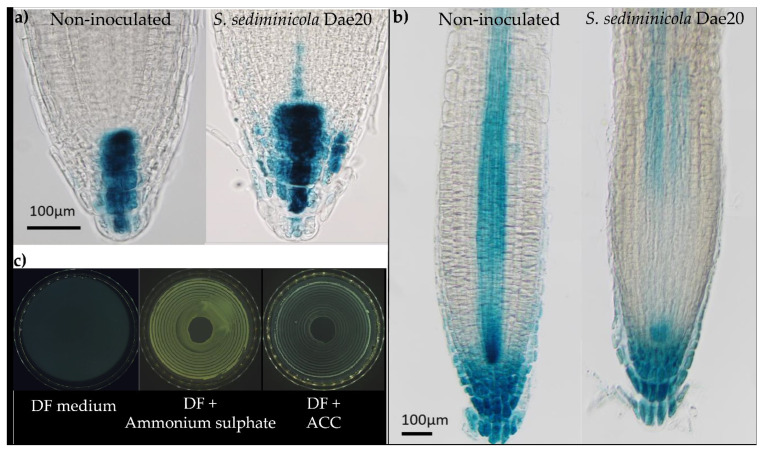
*DR5::GUS* (**a**) and *ARR5::GUS* (**b**) signals in 10-day-old *Arabidopsis* plants non-inoculated or inoculated with *Sphingomonas sediminicola* Dae20. Bars = 100 µm. (**c**) Development of *S. sediminicola* Dae20 on DF medium in a nitrogen-free medium (negative control), DF with ammonium sulphate (positive control) and DF with ACC as sole nitrogen source.

**Figure 4 microorganisms-11-02061-f004:**
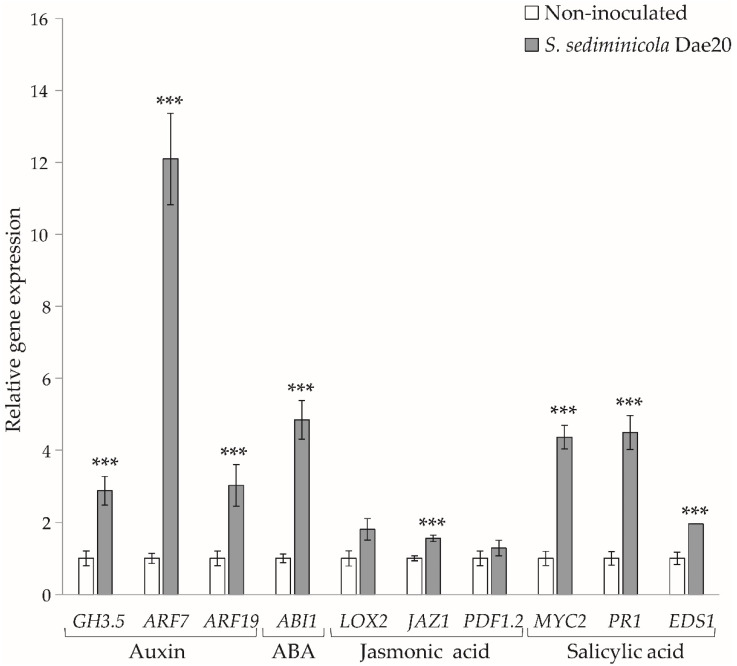
Relative expression of genes involved in auxin, abscisic acid, and salicylic acid pathways when *S. sediminicola* Dae20 was inoculated to *Arabidospsis* seedlings for 10 days. Error bars represent SD. Statistical differences were based on Kruskal–Wallis rank sum tests with Holm’s *p*-adjust. ***, *p* < 0.001.

**Figure 5 microorganisms-11-02061-f005:**
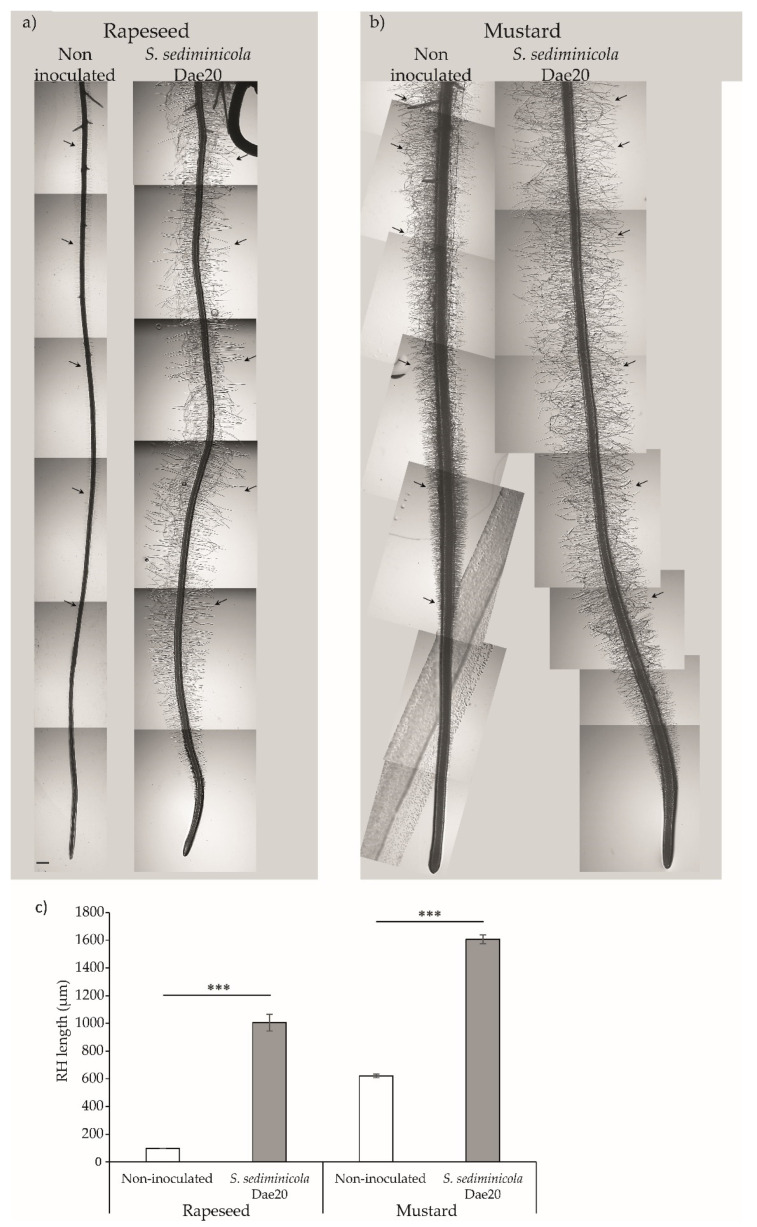
Root of non-inoculated and *Sphingomonas sediminicola* Dae20-inoculated rapeseed (**a**) and (**b**) mustard plants, scale bar = 500 µm. (**c**) Root hair (RH) length of 7-day-old rapeseed or mustard plants non-inoculated (white bar) or inoculated with *S. sediminicola* Dae20 (gray bar). Error bars represent SD. Statistical differences were based on Kruskal–Wallis rank sum tests with Holm’s *p*-adjust. ***, *p* < 0.001.

**Figure 6 microorganisms-11-02061-f006:**
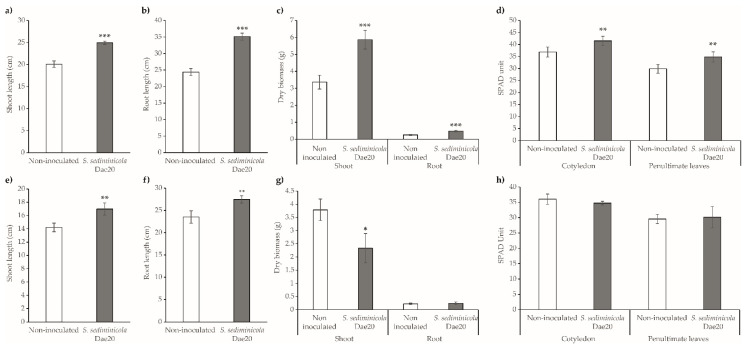
Phenotypic traits of non-inoculated (white bar) and *Sphingomonas sediminicola* Dae20-inoculated (gray bar) mustard (**a**–**d**) and rapeseed (**e**–**h**) plants. (**a**,**e**) Shoot length; (**b**,**f**) root length; (**c**,**d**) shoot and root dry biomass; (**d**,**h**) SPAD values of cotyledons and penultimate leaves. Statistical differences were based on Kruskal–Wallis rank sum tests with Holm’s *p*-adjust. *, *p* < 0.05; **, *p* < 0.01; ***, *p* < 0.001.

**Table 1 microorganisms-11-02061-t001:** Quantification of indole acetic acid (IAA), nitric oxide and siderophores in the supernatant of the free-living *Sphingomonas sediminicola* Dae20 culture.

PGPR Properties	*Sphingomonas sediminicola* Dae20
IAA production (µg/mg dry weight of bacteria)	1.25 ± 0.04
Nitric oxide production (µg/mg dry weight/minutes)	175.81 ± 12.26
Percent siderophore unit (psu)	78.09 ± 4.92

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
