# Peer review of "Sphingomonas sediminicola Dae20 Is a Highly Promising Beneficial Bacteria for Crop Biostimulation Due to Its Positive Effects on Plant Growth and Development"

_microorganisms, 2023, doi:10.3390/microorganisms11082061_

Round 1

Reviewer 1 Report

The study presented results about A novel strain of Sphingomonas sediminicola with capability of promotion of plant growth and development and related mechanisms (production of plant hormones and siderophores). The experiments were prroperly designed and reulsts were properly analyzed and presented. I recommend acceptance of this manuscript for publication. Meanwhile, I have a suggestion. the authors should present the data about the seed yield in the treatments of the bacterium and control. 

Reviewer 2 Report

Dear Authors!

The manuscript is devoted to the actual topic of identifying effective and versatile beneficial bacteria to improve plant nutrition and promote sustainable agricultural practices. The manuscript will certainly arouse the interest of readers, but in order to improve its quality, I propose to make some changes to it.

1. Line 44. The word "bacteria" occurs three times. Replace one of the words with a synonym for -"microorganisms".

2. There is no clear hypothesis and purpose of the study in the Introduction.

3. Section 2.1. Please give a brief description of the strain S. sediminicola Dae 20 (source of isolation, storage location and properties studied earlier). Specify the composition of the R2A medium.

4. The manuscript describes a specific strain, not the S. sediminicola species as a whole, so the strain reference Dae20 should always be given.

5. Lines 113, 142, 172, 183, Fig.1. There are some technical errors in indicating the number of bacteria (5.104 - incorrect, 5 104 or 5 × 104 - correct).

6. Line 118. Briefly describe how the plant material was purified.

7. Fig.1. There is no letter "c" in Fig. 1c. In Fig. 1e, “bc” should be replaced with “c”.

8. Lines 206-207. The text does not match Fig.1g.

9. Fig. S1 should be transferred to the main text of the manuscript. In Fig. S1b, instead of the letters “b”, there should be the letters “a”.

10. Section 3.1. How was it determined that the strain produced VOCs? At the same time, the authors refer to their unpublished manuscript and to the article of other authors on bacteria of the genus Bacillus. Fig. S1a and S1b do not show that primary and lateral root growth was stimulated in the presence of VOCs. How the experiment on growing plants in an atmosphere of VOCs was carried out?

11. Table 1. There is no heading of the first column.

12. Typically, IAA production by strains is measured in 1 ml (µl) of culture liquid. By using μg/mg of dry weight of bacteria as a unit of measurement, the authors made it too difficult to compare the obtained results with the literature data on the synthesis of IAA by other bacteria. It is not clear whether the strain produces a lot or a little IAA.

13. Line 226. There is an error when specifying the figure number.

14. Fig.5d and Fig.5h. Is it possible to compare the SPAD unit in cotyledons and leaves?

15. Lines 295, 303, 329, 339. Species names of strains are not indicated. If the species is not set, then "sp." should be written.

16. The conclusions do not reflect well what is done in this work. They should be more specific. It should not be stated that S. sediminicola Dae20 is capable of causing nodulation in peas, as this plant was not studied in this work.

Minor editing of English language required

Reviewer 3 Report

In the article, the authors study the important issue of the influence of symbiotic bacteria on the growth and development of agricultural plants using the example of the Sphingomonas sediminicola strain. The topic is very relevant due to the global trend towards reducing the use of chemical fertilizers and growth stimulants. The article is well written, has a clear structure. The research methodology is correct. There are only small questions and comments, which are listed below.

General remarks

1) When studying the effect of the S. sediminicola strain on Arabidopsis and other Brassicaceae plants, different concentrations of bacteria were used for each experiment. Please explain the choice of concentrations

2) There are also differences in the set of studied parameters for Arabidopsis and other Brassicaceae. In particular, Arabidopsis does not have data on the relative content of chlorophyll, and other Brassicaceae do not have data on the production of growth hormones. Please explain what causes these differences.

Specific remarks:

1) line 24.79. Remove italics in the title α-Proteobacteria

2) line 49-50. “Symbiotic PGPR, called rhizobia, form a symbiotic relationship…” This part of the sentence is not entirely correct, it is better to write, for example, “Symbiotic PGPR of the genus Rhizobium…etc”

3) line 82. Remove the abbreviation (S.) from the species name Sphingomonas (S.) sediminicola. At the first mention of a species of microorganism, the generic name is written in full, and at subsequent mentions it is abbreviated to the first letter by default.

4) line 92, Materials and Methods. Describe in a little more detail where this strain came from. Isolated from the rhizosphere, collection?

5) line 96. "for 72 h at 30°C with 150 rpm constant shaking" - add on which device the cultivation was carried out (name, manufacturer)

6) line 98-99 "and counted using a 98 Scan® 500 (Interscience)" - add that it was a colony counter

7) line 100. In vitro Arabidopsis assay. Add that there was also a definition of dry biomass, because the results have this data

8) line 206-207 “…and a decrease of 20% for aboveground biomass compared to the non-inoculated plants (Figure 1g)” does not follow from the figure, check the information

9) Figure 1. The graphs and their descriptions are mixed up (in particular, f and g)

10) Figure 3. The words in the x-axis label are merging, you need to make the font smaller

11) line 357. Conclusions. How reasonable is it to speak of an increase in photosynthesis if the content of chlorophyll was measured on mustard and rapeseed (without Arabidopsis), while rapeseed did not show the effect of S. sediminicola strain inoculation on SPAD values?
